# Multi-Agent Big-Data Lambda Architecture Model for E-Commerce Analytics [†]

## Gautam Pal [1,*], Gangmin Li [2] and Katie Atkinson [1]

[1]   Department of Computer Science, The University of Liverpool, Liverpool L69 7ZX, UK;
      K.M.Atkinson@liverpool.ac.uk
[2]   Department of Computer Science, Xi'an Jiaotong-Liverpool University, Wuzhong 215123, China;
      Gangmin.Li@xjtlu.edu.cn
*    Correspondence: gautam.pal@liverpool.ac.uk
[†]   This paper is an extended version of our paper published in the IEEE Second International Conference on
      Data Stream Mining Processing (DSMP), Lviv, Ukraine, 21–25 August 2018.

**Abstract:** We study big-data hybrid-data-processing lambda architecture, which consolidates low-latency real-time frameworks with high-throughput Hadoop-batch frameworks over a massively distributed setup. In particular, real-time and batch-processing engines act as autonomous multi-agent systems in collaboration. We propose a Multi-Agent Lambda Architecture (MALA) for e-commerce data analytics. We address the high-latency problem of Hadoop MapReduce jobs by simultaneous processing at the speed layer to the requests which require a quick turnaround time. At the same time, the batch layer in parallel provides comprehensive coverage of data by intelligent blending of stream and historical data through the weighted voting method. The cold-start problem of streaming services is addressed through the initial offset from historical batch data. Challenges of high-velocity data ingestion is resolved with distributed message queues. A proposed multi-agent decision-maker component is placed at the MALA stack as the gateway of the data pipeline. We prove efficiency of our batch model by implementing an array of features for an e-commerce site. The novelty of the model and its key significance is a scheme for multi-agent interaction between batch and real-time agents to produce deeper insights at low latency and at significantly lower costs. Hence, the proposed system is highly appealing for applications involving big data and caters to high-velocity streaming ingestion and a massive data pool.

**Keywords:** Lambda Architecture; e-commerce analytics; real-time data analytics; real-time data ingestion; real-time machine leaning; recommender engine; online k-means clustering

## 1. Introduction

Big-data Lambda Architecture (LA) attempts to balance high-throughput MapReduce frameworks with low-latency real-time processing. Apache Hadoop is the de facto standard batch-processing system used to provide high-throughput, comprehensive and more accurate views of historical data. However, it suffers from several challenges such as high latency, larger storage, and bigger cluster requirements.

This paper sought to shed light on an optimized framework for the e-commerce domain through LA. The batch and real-time components of LA are two autonomous agents which collaborate at need. We introduce novel concepts of Multi-Agent Lambda Architecture (MALA), bringing unprecedented optimization through blending low-latency stream processing with comprehensive batch processing, which fits the analytical challenges of the e-commerce domain.

LA is a positive new direction towards the next generation of big-data analytics and has started to witness wide industry adaptation in recent years. However, research efforts towards LA are still in their infancy. We claim this work is the first general-purpose and comprehensive approach in combination with big data and MALA designed for hybrid mixed learning of batch and stream data.

The main contributions of the paper are as follows:

*High Responsive Streaming Layer:* We prove that within the MALA framework, streaming data can serve several functionalities in the speed layer. The novelty in the approach is that a big-data multi-agent scheme enables graceful interaction of the stream data with historical batch data at the initiation and thereafter in an autonomous way.

- Detect the outliers through hybrid clustering algorithm.
- Predictive analytics through mixed processing to forecast network traffic loads for the e-commerce portal.
- Identify user buying patterns through the complete information trail from viewing to buying an item.
- Ingested clickstream data simultaneously gets cleansed and processed to serve the user interface (UI) layer and display the list of recently viewed items by each user to enhance the overall shopping experience.
- List of *Recently Viewed Items* by a user determines the final order of recommended products through an algorithm built upon weighted average between historical batch data and live streaming data.

*Recommender Service at Batch Layer:* As proof to the MALA batch module, we demonstrate a multi-agent collaboration approach which blends a real-time stream with historical batch data to produce recommendations with outstanding accuracy. For brevity, this work does not elaborate on its core recommendation engine. Our primary focus is to demonstrate the influence of stream data over batch model, which is as follows:

- This paper redefines the item-to-item relationship by putting more relevance on current trends than the historical data.
- We use a hybrid approach to mix results from Collaborative Filtering with content-based filtering on item similarity.
- Based on the above studies, we present a novel architecture of an end-to-end recommender system with a host of online and offline big-data ecosystem tools and their correlation as multi-agent interaction model.

*Big-Data Architecture:* We analyze user clickstream data. Clickstream is intensely data exhaustive compared to final purchase data. We propose a robust big-data infrastructure to support the enormous storage and processing requirements.

### 1.1. Organization of this paper

The organization of this paper as follows: in Sections 3 and 4 we introduce the LA and MALA. Sections 5 and 6 highlights the hybrid learning architecture through stream and batch modules. Experimental results are presented in Section 7. The paper concludes with several remarks in Section 8.

### 1.2. Key Findings

Smart collaboration between batch and real-time agents delivers deeper insights at low latency. The model gets more precise over time with incremental learning. It produces significant infrastructure cost savings through smaller cluster requirements and improves training time, enabling quick adaptability to learn newer changes faster.

## 2. Background and Related Work

Several architectures were introduced for LA, involving applications dealing with the high velocity of data ingestion. Scalable stream processing was started as a new category of open-source projects by Twitter's Nathan Marz [1], who also designed the generic model for LA.

In their work, Yamato et al. [2] illustrated a model for real-time predictive maintenance on an IoT deployment for NTT DOCOMO. The model analyzes the sensor stream data for anomalies at the speed layer, which requires prompt actions. In the batch layer, raw data are sent to the cloud and stored in DB with low-cost methods such as night transfers. A maintenance application predicts the failure to analyze data in detail, which does not need prompt actions.

In the area of hybrid active learning, Kim et al. [3] extended ideas such as online-learning algorithms and batch processing to come up with a model which integrates pool-based and stream-based sampling strategies for active learning to address scenarios where concept drift is prevalent, and labeling is asynchronous.

A few notable domain application areas have emerged through these for LA such as telecom, e-commerce, social networking, and manufacturing [4].

Lee et al. [5] implemented the LA model for a restaurant recommender system. Their work builds upon several open-source pieces of software such as Apache Mesos, Kafka, and Spark. At the speed layer, the pipeline processes the incoming stream data to compute user rating, while batch layer is designed to process the large data offline and execute complex machine learning algorithm using Spark.

Apache Storm and Spark are two prominent stream-processing platforms for big data. Batyunk et al. [6] show a LA implementation for processing streaming data from social networks using Apache Storm to perform the task to build up a predictive model of trends based on data streams from GitHub and Twitter.

Hanif et al. [7] proposed an adaptive watermarking and dynamic buffering timeout mechanism for Apache Flink, which is designed to increase overall throughput by making he watermarks of the system adaptive according to the input workload.

In real-world applications, it is common to have access to implicit feedback in the form of views, clicks, purchases, likes, shares, comments etc., Still a overwhelming majority of existing research just focuses on users' explicit feedback (ratings). In this paper Y. Hu et al. [8] proposed an implicit feedback-based recommender system to improve customer experience through personalized recommendations depending on prior implicit feedback. In our work, we do not access ratings provided by the users and do not access user preferences specified during registration with the portal. Implicit feedback is purely counted in terms of the item viewed by users.

The majority of the existing recommender approaches ignored contextual information such as data age, place etc. For example, in the case of online shopping portals, buying pattern is largely influenced by geographic location such as colder states vs warmer states, remote areas vs cities, browsing from desktop vs mobile etc.. In other words, recommender systems deal with two types of entities—users and items—but do not put them into a context when providing recommendations. Context here is a multifaceted concept that has been studied across various research disciplines, including computer science, cognitive science, linguistics, philosophy, psychology etc., [9–13]. Our work takes an approach to consider *context information* as a latent factor to provide more meaningful recommendations depicted as flows: Contextual $RecommendationSystem = Users \times Items \times Context \rightarrow$ Implicit Feedback (clicks)-Based Recommender System (IFBRS). For example, time of browsing is extracted, as a context provides *data age*. IFBRS prioritized recent data over the historical data in the final recommendation ordering.

## 3. Big-Data Lambda Architecture

In this section, we present a novel hybrid processing and learning approach through big-data LA. LA [14] combines both the batch and stream-processing approaches. As depicted in Figure 1, the full

dataset ingested to the system is moved to both batch and stream layers for processing. The stream layer serves only low-latency queries. Data gets merged for type of data mining, which requires historical data. The architecture is classified into three layers:

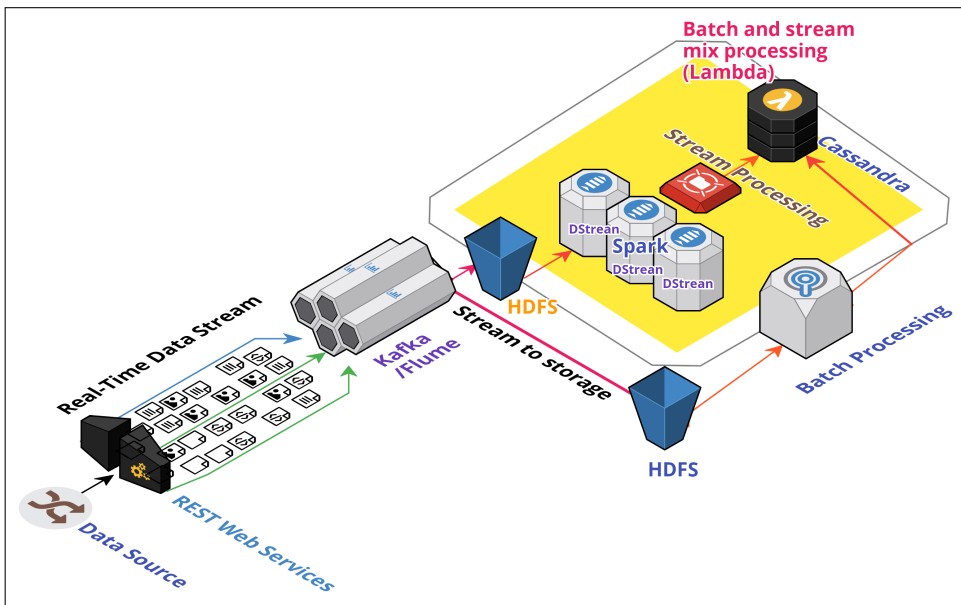

**Figure 1.** Lambda architecture (LA), which combines low-latency real-time frameworks with high-throughput Hadoop-batch framework over a massively distributed setup. Observe that while data is being processed in real time as Spark DStreams, simultaneous batch processing takes place through the stored data from Hadoop Distributed File System (HDFS). Cassandra stores the combined view from batch and stream.

### 3.1. Batch layer

This is a high-throughput/high-latency option. Processing duration varies from a few minutes to hours. Data is ingested in large batches at a certain schedule, reports are generated, users check the same reports until the next data load occurs. Frameworks and solutions such as Hadoop MapReduce, Spark core, Spark SQL, GraphX, and MLLib are the widely adapted big-data tools using batch mode. Batch schedulers include Apache Oozie, Spring Batch, and Unix Corn which, invoke the processing at a periodic intervals.

### 3.2. Speed Layer

This is a continuous non-blocking option which is for low-latency messaging and event process, responding to user requests in real time or near real time. Most operations on streams are windowed operations operating on slices of time such as moving averages for stock process every hour, top products sold this week, etc.. Popular choices for stream-processing tools include Apache Kafka, Apache Flume, Apache Storm, Spark Streaming, Apache Flink, Amazon Kinesis etc., [15].

### 3.3. Serving Layer

Serving layer consolidates batch and real-time views into one and retrieves results in real time for on-demand queries over the entire dataset. It provides low-latency access on the full dataset. Our approach considers real-time and batch components as autonomous, self-organizing, collaborative multi-agent systems.

## 4. Multi-Agent Lambda Architecture(MALA)

MALA is a consolidation framework for stream and batch modules based on the fundamentals of LA. We design this as an extension to LA and our primary continuation in this paper. The framework enables collaborative, accumulative learning through big-data tools and APIs as shown in Figure 2. Streaming and batch components act as a cooperative autonomous multi-agent systems. See Figure 3.

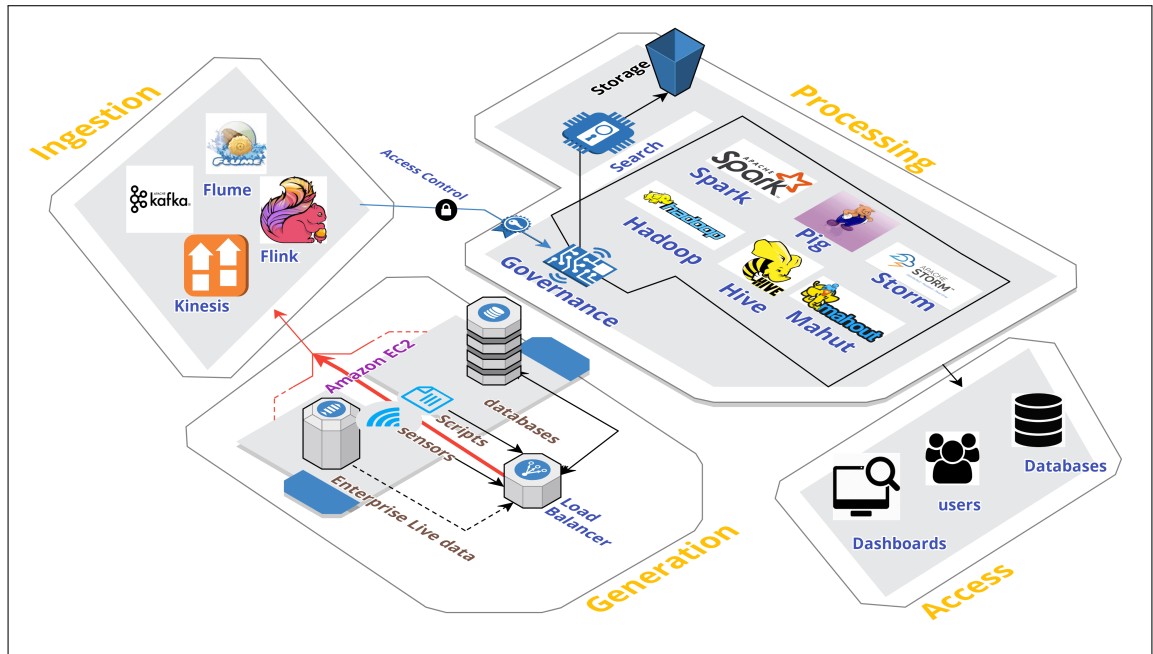

**Figure 2.** Typical four-layered big-data architecture: ingestion, processing, storage, and visualization. The proposed framework combines both batch and stream-processing frameworks. Not necessarily every application implements all the components together.

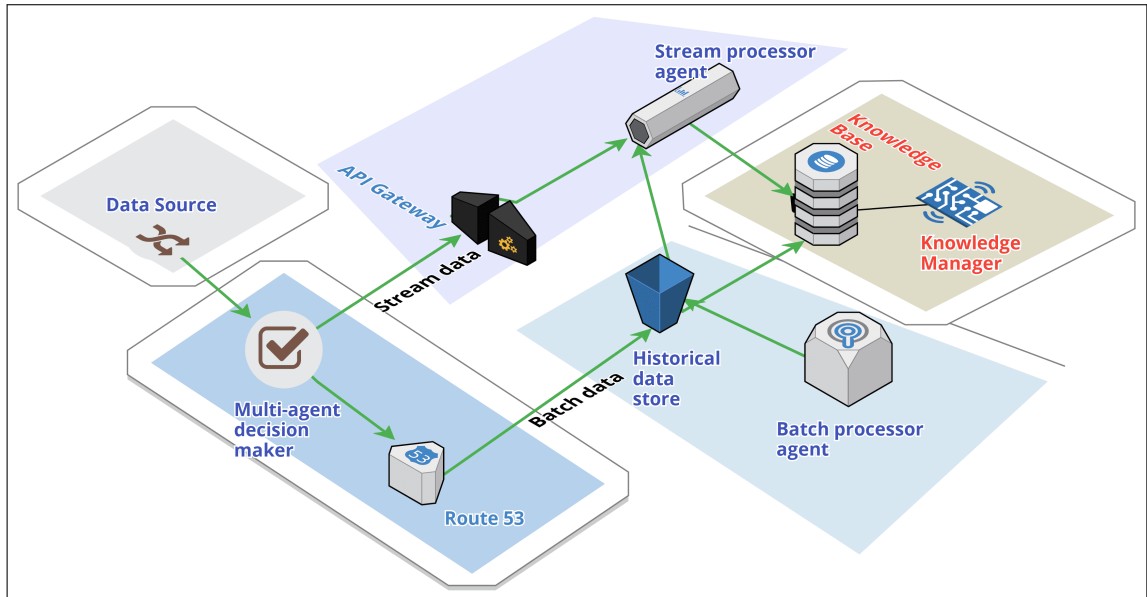

**Figure 3.** Multi-Agent Lambda Architecture (MALA). Three components of MALA are: historical data store, stream processor, knowledge miner. Real-time and batch layers act as autonomous multi-agent systems in collaboration. A multi-agent decision-maker component is placed in the MALA stack at the gateway of the data pipeline for further processing.

*4.1. Historical Data Store*

A large volume of static data pool, all at once, at a periodic interval, gets ingested, processed and written back using distributed frameworks such as Apache Hadoop and Spark. Input data is stored over time in a distributed file system such as Hadoop Distributed File System (HDFS), NoSQL databases, or Amazon Simple Storage Service (S3). The model is trained against the entire data pool. In batch mode, response time is not a big constraint but rather design is inclined towards a comprehensive coverage of the data. Historical Data Store (HDS) persists in the trained models.

*4.2. Stream Processor*

MALA updates its model with each new wave of incoming data. The streaming model initializes itself with *saved learning* from batch by loading the trained model from persistent distributed storage into distributed memory. Continuous data stream updates its model incrementally. The training time largely depends on mini-batch data size and window length. Duration can vary from a few milliseconds to minutes. We use Apache Spark Streaming to create in-memory DStreams from the stored model in HDFS produced by the batch jobs. After each iteration of re-training, the updated model is persisted into memory and disk. Since the model keeps growing, the most recent data is cached into distributed memory and the remainder goes to disk. Amount of distributed memory size is configurable through the Spark configuration file. Stream processing includes filtration of rows and transforming the ingested flow into structured data.

*4.3. Knowledge Miner and Knowledge Base*

Knowledge Miner (KM) consolidates the past learning with recent stream updates into Knowledge Base (KB). KM is responsible for filtration (eliminating unwanted and faulty records), knowledge aggregation and data governance for monitoring and reporting purposes. Filtration and aggregation logic is ad-hoc, developed for a specific case study. Refer to Section 5.1 for a hybrid learning approach powered by KM.

## 5. MALA Streaming Framework

This section discusses the streaming module of MALA. The framework turns iterative learning of streaming models into *lifelong learning machines* [16,17]. It also removes any cold-start situations to begin with. The streaming model initializes itself with *saved learning* from batch by loading the trained model from the HDFS into distributed memory. MALA updates its streaming model incrementally and continuously on each new wave of incoming data as shown in Figure 4. The framework further allows the merging of large static historical data pool with the latest and most updated streaming model.

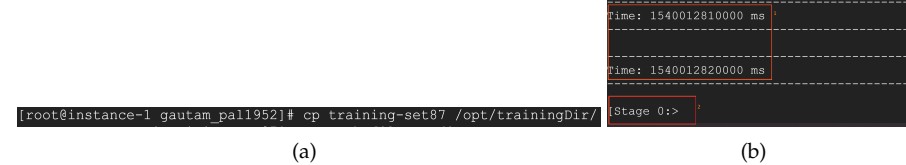

| | |
|---|---|
| (a) | (b) |

**Figure 4.** Screenshot for incremental streaming learning. MALA initializes with historical batch data and updates its streaming model incrementally and continuously on each new wave of incoming data. Model runs indefinitely at a 10-second window interval. (**a**) The training folder is updated incrementally with new data; (**b**) The model is retrained simultaneously with a small amount of new data as they appear.

*5.1. Hybrid Learning in MALA*

During the training, feature vectors initialize with historical batch data and write them into HDFS. A long-running batch schedule trains the vector and the response is stored to HDFS. One specific

problem with this setting is that the full dataset may not be available at the beginning of training schedule. The other key challenge is that the vectors can get updated with new raw events over time. A stream processor loads the model from HDFS and updates on top of it in an incremental fashion. The correlation $r_{ui}$ between user and item in the shopping portal is expressed as:

$$f(r_{ui}) = v_u v_i \tag{1}$$

$v_u$ and $v_i$ are the feature vector for user and item, respectively. This approach has the limitation of responsiveness with a large dimension of user and item feature vectors coupled with a huge volume of data pool. The other challenge is that the vectors can get updated with new raw events arriving over time. Therefore, there arises a constant need to revise the feature vectors, which leads us to an iterative approach. The vectors will initialize with historical batch data and write them into HDFS. A stream processor loads the model from HDFS and builds on top of it in an incremental fashion. Therefore, we update Equation (1) to accommodate the incremental (delta) learning as follows:

$$f(r_{ui}) = v_u v_i + \delta_u \delta_i \tag{2}$$

We learn $v_u$, $v_i$ offline and $\delta_u$, $\delta_i$ at an iterative fashion with small amount of streaming data. $\delta_u$ and $\delta_i$ are the most current updates or fresh new rows of data. Dimensionality of $\delta_u$ is the same as $v_u$ or dimensionality of $\delta_i$ is same as $v_i$ but the number of rows or overall data volume is just the volume of data collected at a streaming window length of a few minutes and therefore not so huge. The design enables the model to keep updated at low latency on a small amount of incremental data. See Algirithm 1.

Agarwal et al. [18] proposed a *dimension reduction* approach for vectors. Therefore, $\delta_i$ or $\delta_j$ need not to be of the same dimension regarding user or item vector. Instead, only the modified columns update the training model through the online process. This approach reduces the dimension of streaming learning data and the online-learning time, since the online model only needs to learn the *correction* over the batch offset. However, a search-and-compare for the rows and columns in it which requires an update and merge with the batch proves to be expensive on a large volume of data pool. Hence, our approach to *retrain* the model on a small window interval of data pool through distributed in-memory processing of Apache Spark yields better outcome for response time. Dimension reduction however can improve streaming learning with an efficient search and merge technique.

---

**Algorithm 1** Lifelong Hybrid Learning Algorithm

---

**Input:** historical batch datastore $d_b$, dataset as collection of streaming records since last window
　　$d_s = \sum_{i=1}^n d_i$, Kafka topic $k_t$, DB server address $d_a$,
**Output:** : Updated knowledge base kb
　1: kb ← BatchComputationRule($d_b$)
　2: call SubscribeKafka();
　3: **for** windowed dataset $d_s$ **do**

　4:　　Spark consumes Kafka queue
　5:　　$C_s$ ← StreamComputationRule($d_s$)
　6:　　**B** ← Update($C_b, C_s$) //update batch with stream
　7:　　**return** kb //kb is persisted into distributed memory and HDFS in parts
　8: **end for**

---

*5.2. MALA K-Means Clustering*

MALA k-means Clustering implements the logic by Equation (2). The batch version of k-means algorithm provides an offset (or initial point) for the streaming learning to update the model iteratively. The basic idea for an online version of k-means clustering is to divide the data stream into mini-batch windows and to incorporate knowledge learned in the previous window into the following ones. Hence, in streaming k-means clustering, the model is updated with each rolling window based on a

combination of cluster centers computed from the preceding mini-batches and the current mini-batch. The streaming algorithm is adapted from the most recent release of Apache Spark [19]. The algorithm starts with initializing data points to their closest clusters. For every iteration, with a new data arrival, we evaluate new cluster centers, then update each cluster using:

$$c_{t+1} = \frac{c_t\, n_t\alpha + x_t m_t}{n_t\alpha + m_t} \tag{3}$$

$$n_{t+1} = n_t + m_t \tag{4}$$

where, $n_t$ is the past data points and $c_t$ is the past cluster center. $m_t$ is the latest data points and $x_t$ is the latest cluster center. $\alpha$ is the decay factor. With $\alpha = 0$, only the most recent dataset is used; for $\alpha = 1$, all data is used from beginning. Apache Spark MLLib [19] libraries include a streaming version k-means clustering.

In the context of e-commerce, we use streaming k-means clustering to detect any outlier for any intrusion or other system failure which requires a quick turnaround time. Refer to Section 7 for experimental results.

### 5.3. Recently Viewed Products

We have so far discussed several functionalities at the speed layer which demand a quick response time. To this end, we analyze clickstream data at the Spark streaming layer to display *recently viewed products* to each user, which influences the analytics at the batch layer.

Say a user $u_i$ successively viewed items a, b, c in the same session at time $t_1$, $t_2$, $t_3$ where $t_3 > t_2 > t_1$. Recently viewed products for user $u_i$ would be the reverse chronologically ordered list of products: c, b, a. The main challenge here is identifying each user uniquely even when they are not logged in. In the absence of user ID, we create context ID for each user click datum. User *context* is derived from the session object which is associated with each time a user newly opens the e-commerce website. Each time, it is a new session and a new context to be captured. The context is a uniquely derived object created from a session object created at the JavaScript layer. Context ID would then be appended to each user click datum.

Furthermore, a recently viewed product list has an influence at the batch layer to determine the final order of recommended products for each user, which is discussed in the following section.

## 6. MALA Batch Frameworks

The general notion of MALA is for simultaneous execution of batch and real-time view, which often mixed with each another to present a more comprehensive, accurate view. Preceding sections presented a number of functionalities under the real-time view pertaining to the e-commerce domain. In this section, we present a very brief overview of a novel architecture of an e-commerce recommender system under batch settings which uses MALA as its central concept. *Recently viewed items* by each user are computed under streaming setting and will have an impact on the final recommendation order, which is computed under batch settings. Also, the recommender system presented here does not enforce users to log in to see recommendations; rather it identifies each user uniquely from the application server-generated context ID, through users' own click data.

This work refrains from divulging in detail the design of the recommender system and focuses on the impact of stream data into batch modelling through the proposed algorithm. Readers may refer to our previous work [20] for detailed coverage on building recommender engine.

### Computing Recommended items

A weighted hybridization strategy combines the recommendations of two or more latent factors by computing the weighted sums of their scores. This step is performed once items are filtered out

based on similarity count by Equation (5). The overall recommendation score R can be calculated for a user by the Equation (6):

Different latent factors are:

- Co-occurrence count
- User location
- User preferences
- Timestamp of the click data in the recently viewed table.

To filter out unrelated item pairs, we compute cosine similarity between items by comparing identical features. Calculate the item similarity strength of product A and B at a scale of 0 to 1 by Equation (5) and Algorithm 2:

$$S_{A,B} = cos(\theta) = \frac{(A \times B)}{(|A||B|)} = \frac{\sum_{i=1}^{n} A_i \times B_i}{\sum_{i=1}^{n} \sqrt{A_i^2} \times \sqrt{B_i^2}} \tag{5}$$

$S_{A,B}$ is the similarity between product A and B, $cos(\theta)$ is the angle between edges A and B. Once the items are filtered out based on similarity count, overall recommendation score R is calculated for a user by Equation (6).

$$R = \prod_{k=1}^{n} \overline{\omega}_K \tag{6}$$

$\overline{\omega}_K$ is the normalized form of each latent factor of weight $\omega_k$ The normalization equation is as follows:

$$\overline{\omega}_K = \frac{\omega_k}{max_k(\omega_k)} \tag{7}$$

where $\omega_k$ is the view count of the $k^{th}$ product and $max_k(\omega_k)$ is the maximum view count across all categories which user viewed. For our experimental setting, view count is kept for a boundary period of two weeks. $\overline{\omega}_K$ is the normalized for each latent factor of weight $\omega_k$

See Algorithm 2 for the complete flow of the recommender engine. Readers may refer to [20] for the detailed steps for the recommendation engine.

---

**Algorithm 2** Implicit Feedback-Based Recommender System

---

**Input:** Customer clickstream data, time-decay-factor
**Output:** : top n recommended product list

*initialize item_timestamp_weight=x, item_decay_factor=y*
1: **for** each customer C who viewed item $I_i$ **do**
2:    Record all the items recently viewed $I_i$ ($i$ = 1 to $n$) in
   reverse chronological order
3:    **for** each recently viewed item $I_i$ **do**
4:       Record customer C also viewed related item $RI_i$
5:       **for** each item pair $I_i$, $RI_i$ **do**
6:          Filter pair $I_i$, $RI_i$ based on cosine similarity
7:          Compute the co-occurrence count between pair $I_i$, $RI_i$
8:          Apply location factor
9:          Apply user preferences
10:         Apply Equation (6) to decide overall score
11:       **end for**
12:      *item_timestamp_weight=item_timestamp_weight − time_decay_factor*
13:    **end for**
14:    Compute and the final recommendation order of $I_i$
15: **end for**
16: **return** top n recommended product list

---

## 7. Experiments

A real-world e-commerce dataset is obtained from e-commerce streaming data generation app OpsDataGen.spl [21] and data.world [22] consisting of data from Indian e-commerce retailer Flipcart.

This section provides the proofs through comprehensive experiments carried out in Amazon Cloud Services (AWS). In each subsection, we present the experimental outcomes for the topics discussed in the preceding sections.

### 7.1. MALA Predictive Modelling

This section demonstrates results achieved through MALA collaborative mix processing framework. The framework uses the batch and streaming linear regression APIs from spark.mlLib libraries. The following results are presented here:

- Forecast chart for network traffic load for e-commerce portal. See Figure 5.
- Punch card view of user behavior in the e-commerce portal grouped by day of the week. See Figure 6.
- Outlier chart for unusual buying patterns. See Figure 7.
- Rolling hourly prediction count of number of checkouts. See Table 1.

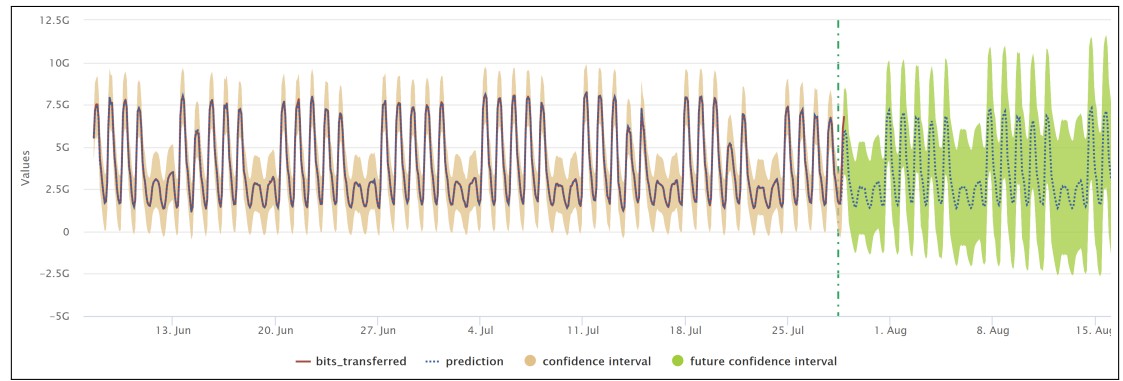

**Figure 5.** Forecast chart for network traffic load for e-commerce portal. Vertical green dotted line differentiates past and future data projections. Confidence interval is 95%.

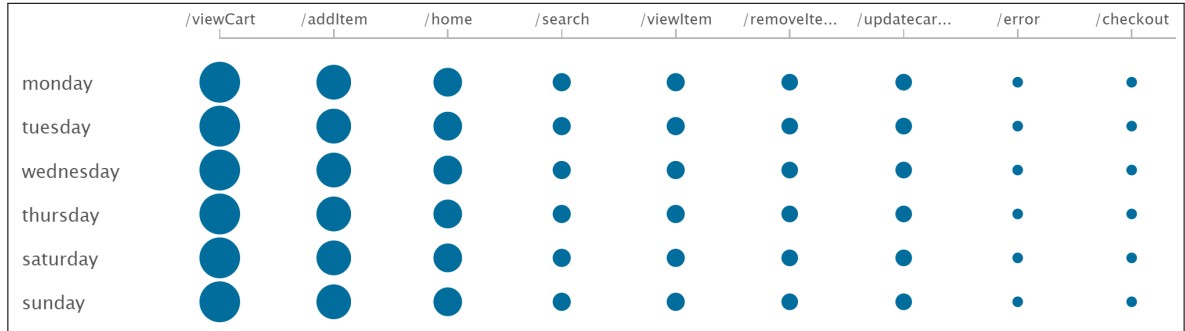

**Figure 6.** Punch card view of user behavior in the e-commerce portal grouped by day of the week. It also reveals the chart conversion from item added to cart to final checkouts. Only a tiny percentage of cart additions converts to checkouts. With MALA using a moving average of one week, graph can be updated at the end of each week and predict future behavior.

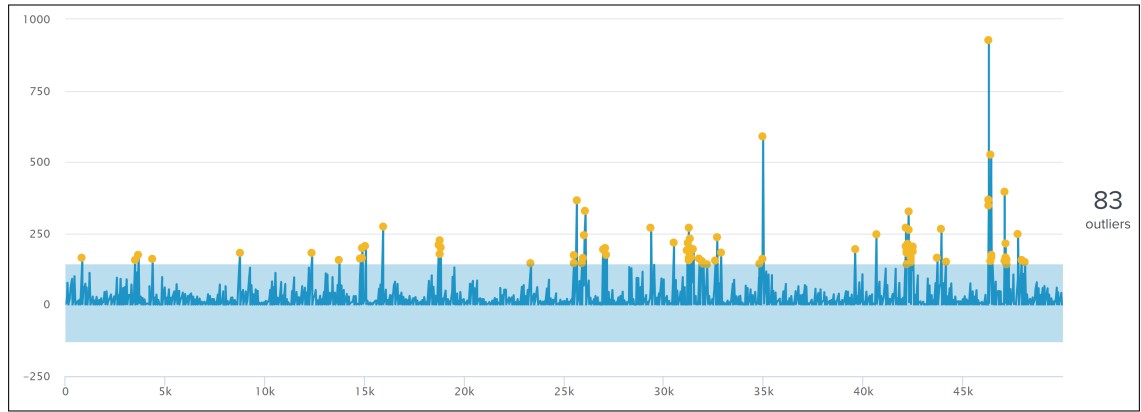

**Figure 7.** Outliers chart for unusual buying patterns. The x and y axes mark the data volume and quantity of purchases. The algorithm finds the outlier for unusually high or low quantity of purchases through the mix of stream and batch k-means clustering methods. Yellow dots mark the outliers.

**Table 1.** Moving hourly prediction count of number of checkouts.

| Time Hour | Actual Count | Lower95 Prediction Count | Upper95 Prediction Count |
|---|---|---|---|
| 0:00 | 106 | 105.0630306 | 106.9369694 |
| 1:00 | 208 | 207.0629725 | 208.9370275 |
| 2:00 | 206 | 205.061734 | 206.9357845 |
| 3:00 | 186 | 185.0629788 | 186.9370212 |
| 4:00 | 206 | 205.0631755 | 206.9372127 |
| 5:00 | 208 | 207.0636578 | 208.9376911 |
| 6:00 | 206 | 205.064078 | 206.9381085 |
| 7:00 | 186 | 185.0649051 | 186.9389334 |
| 8:00 | 206 | 205.064087 | 206.9381136 |
| 9:00 | 206 | 205.0643379 | 206.938363 |
| 10:00 | 188 | 187.0652488 | 188.9392727 |
| 11:00 | 224 | 223.0635381 | 224.937561 |
| 12:00 | 188 | 187.065613 | 188.939635 |
| 13:00 | 206 | 205.0645522 | 206.9385736 |
| 14:00 | 206 | 205.0646471 | 206.9386678 |
| 15:00 | 206 | 205.0647169 | 206.938737 |
| 16:00 | 190 | 189.0656084 | 190.9396281 |
| 17:00 | 204 | 203.0646925 | 204.9387117 |
| 18:00 | 208 | 207.0645059 | 208.9385247 |

The table displays the prediction for moving average of one hour. Confidence interval is 95%. To counter the cold-start situations, MALA uses the historical batch data as initial offset for stream engine to update incrementally.

## 7.2. Load Test

This section presents load test results for Cassandra DB in the architecture. Server setup is shown in the Table 2 and database setup is presented in Table 3. Refer to Figures 8 and 9 for test results obtained through Datastax OPSCenter. The results assert the scalability of the system under stress.

**Table 2.** AWS Instance Types.

| Instance Type | Instance Count | vCPUs | Memory | Instance Storage | EBS Optimized Bandwidth |
|---|---|---|---|---|---|
| m1.large | 3 | 2 | 7.5 GB | 500 GB | Moderate |

**Table 3.** Cassandra Setup.

| Cassandra Vendor | Number of Records | Replication Factor |
|---|---|---|
| DSE | 6 Million writes | 3 |

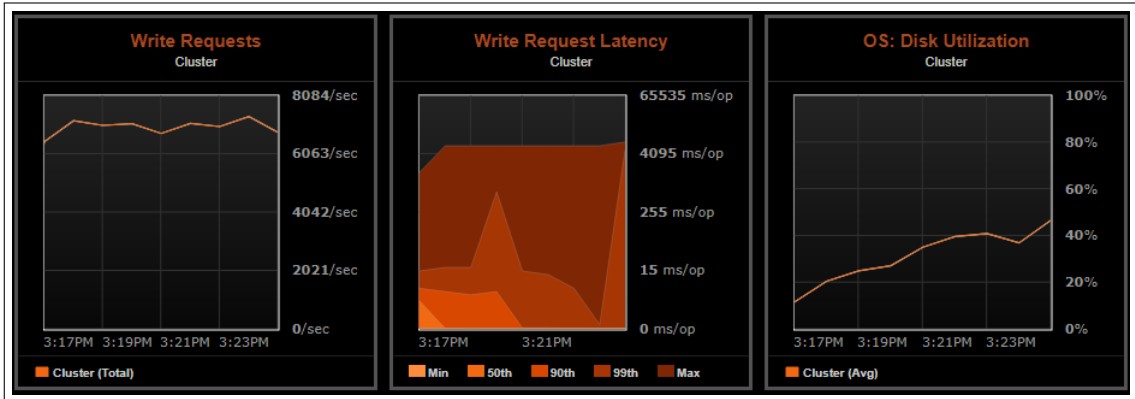

**Figure 8.** Statistics shown for write requests, write request latency, OS disk use. Write requests: the count of writes per second on the coordinator system. Monitoring the number of requests over time exposes system write load and usage patterns. Write request latency: The 90th and 99th percentiles, min, median, max of a client node write. The period initiated with a node receives a client write request and ends with the node responding back to the client. Considering the consistency setting and replication factor, this includes the network delay from writing to the replicas. OS disk use: CPU time used by disk I/O.

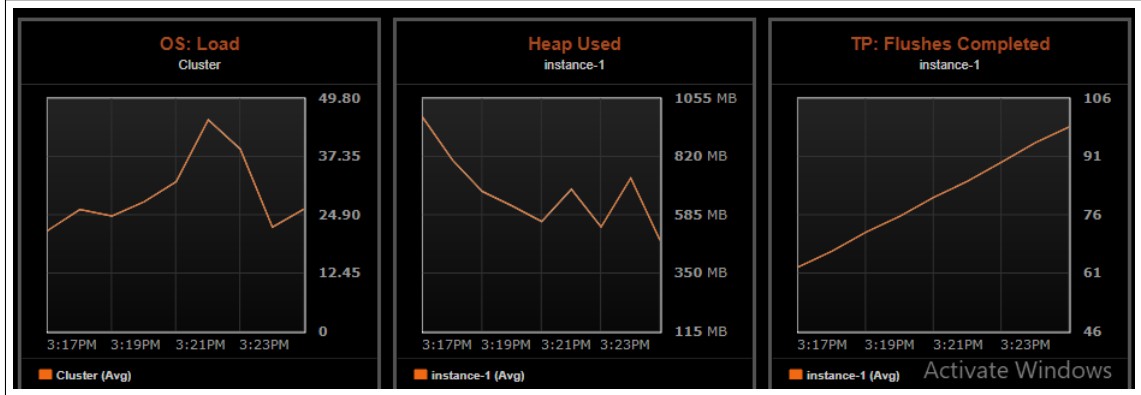

**Figure 9.** Statistics shown for OS load, Heap used, TP flushes completed. OS load: Operating system load average. One-minute value parsed from /proc/loadavg on Linux systems. Heap used: Average amount of Java heap memory used. TP flushes completed: Number of memtables flushed to disk since the nodes start.

## 8. Results Summary and Discussion

### 8.1. Comparing efficiency of MALA Hybrid Learning

MALA is a collaborative consolidation framework for stream and batch data. Its real-time and batch agents use Spark MlLib APIs. MALA offers the following key advantages over standalone batch and stream frameworks:

*Smart Collaboration:* Interactive real-time and batch agents can collaborate, retrain, consolidate, and exchange learning from individual domains to provide deeper analytical insights. The experimental results illustrate the oracle performance significantly by low-latency processing at the speed layer and providing comprehensive coverage at the batch layer. This is the key contribution for the model and is highly appealing for the e-commerce administrators and managers looking for the optimization in analytical insights.

*Faster Training Time, Quick Adaptability, Horizontal Scalability, Infrastructure Cost Savings:* When the application cannot wait until the entire dataset is collected, an iterative approach can help train the model with the available dataset. Training set can adapt much faster to match with the new updates in

a raw dataset by re-training iteratively. The load test results assert the excellent scalability of the system under different stress conditions. When it is not feasible to train the entire set due to large-size cluster requirements, the incremental mode allows the model to be trained iteratively on a small amount of data at a relatively modest infrastructure setting with commodity hardware.

*Comparing Batch and Real-Time Module:* One of the intriguing aspects of the LA model is measuring the quality of real-time algorithm with respect to its batch counterpart. A streaming algorithm is called $\alpha$ competitive if there exists positive constants $\alpha$ and $\gamma$ such that:

$$v_i \leq \alpha v_b + \gamma \tag{8}$$

$v_i$ is the cost for the real-time algorithm $v_b$ is the cost for the batch algorithm

From Equation (8) we can derive that an $\alpha$ competitive streaming algorithm A has cost no worse than $\alpha$ times that of the optimal batch algorithm ($v_b$) plus some initial advantages ($\gamma$) given to the optimal algorithm based on the problem setup [23].

### 8.2. Limitations

The model has the limitation of keeping the code base in the stream and batch layers which produces the same results. The stream layer often performs additional functionalities (such as recently viewed products in the recommender system) but it still redoes all the major tasks by the batch layer. This leads to a typical sync problem, wherein any changes made in one layer need to be mirrored to the other. Also, having trained on the same volume of dataset, the accuracy of the proposed model is no better compared to the batch-only model for certain functionalities, which does not involve collaboration between two modules. Several e-commerce analytics, especially those not involving large data chunks, may find the architecture over-complex.

### 9. Conclusions

We studied big-data hybrid-collaborative MALA for the e-commerce domain. The aim was to provide an approach for an intelligent blend of historical batch data with a real-time stream. MALA allows low-latency processing through real-time frameworks while simultaneously providing comprehensive and accurate coverage though long-running batch frameworks. Graceful interaction of the stream with historical batch data provides deeper insights at low latency. Proposed frameworks provide a solution to the existing imitations of the Hadoop framework, which is inherently batch oriented. High-velocity data integration with source to the processing engine is managed through distributed message queues such as Kafka and Flume. Within the MALA framework for the e-commerce domain, the streaming data can serve a number of functionalities in the speed layer, including predictive analytics, identify buying patterns and detecting outliers. Within the scope of the batch layer, we discuss a novel approach for the *IFBRS*. The novelty of this approach is in the use of the contextual parameter with the collaboration of the multi-agent system for historical batch data and near real time *recently viewed* data to predict recommendations accurately. Distinct advantages of the proposed model over the Hadoop MR and Spark ML APIs are in terms of response time, reduced training time, handling cold-start situations and significant infrastructure cost savings. The limitations include inherent design complexity, sync issues, and functional redundancies of the two layers.

**Author Contributions:** G.P., G.L., K.A. conceived and completed formal analysis; G.P. performed methodology, data curation, experiments, visualization and wrote original draft; G.L., K.A. contributed towards supervision, review, editing and fund acquisition.

**Funding:** This research was funded by Accenture Technology Labs, Beijing, China. Grant number RDF 15-02-35.

**Acknowledgments:** The applications were freely deployed on the cloud infrastructure provided by Research Institute of Big Data Analytics (RIBDA), Xi'an Jiaotong-Liverpool University, Suzhou, China.

**Conflicts of Interest:** The authors declare no conflict of interest.

## Abbreviations

The following abbreviations are used in this manuscript:

LA      Lambda Architecture
MALA    Multi-Agent Lambda Architecture
KM      Knowledge Miner
KB      Knowledge Base
MADM    Multi-Agent Decision Making
RDF     Resource Description Framework

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
