# Peer review of "Multi-Agent Big-Data Lambda Architecture Model for E-Commerce Analytics†"

_data_

Round 1
Reviewer 1 Report
This paper addresses a multi-agent big data lambda architecture model for e-commerce analytics. The topic is very interesting and it’s very beneficial for retail practitioners. Presently, the manuscript is too long and has serious structural issues. At times, it is difficult to grasp the real meaning of the sentence. Though, I have some comments which I would like to be addressed before the acceptance of the paper.
Major Comments:
1. Please add the novelty of your work in abstract.
2. I recommend the authors to follow the MDPI’s proper Introduction format. The introduction should briefly place the study in a broad context and highlight why it is important. It should define the purpose of the work and its significance. Finally, briefly mention the main aim of the work and highlight the main conclusions. Keep the introduction comprehensible to scientists working outside the topic of the paper. Currently, authors add a subsection about the “contributions” of the study. I would rather suggest to merge it as a paragraphs and avoid the subsections.
3. Please provide the summary of the most important results in the last paragraph of the introduction section.
4. I would suggest authors to move the section 2.1. “Organization of this paper” into the introduction section.
5. The Figures 2, 12, 13, 14, 17 and Table 7, 8 are not mentioned in the manuscript. Please explain the figures and tables with a proper evidence grabbed from the graphical results with reason.
6. Section 5.2 describes the “Capturing Stream Data”. The authors describe three approaches and numbered them twice (in the start and end). For example, line 207 “1. Approach 1” and so on. Please modify accordingly.
7. Page 7, line 208 explains the disadvantage of Approach 1. While, in case of Approach 2 and Approach 3 advantages are explained. The authors should explain in a streamline manner rather than jumping from disadvantage to advantages. Furthermore, avoid new bullets in this section and just explain in one or two lines the advantages or disadvantage.
8. I would suggest authors to modify and merge the discussion and conclusion section as one. Additionally, explain how the study findings will help the managers in E-commerce context by adding some theoretical and managerial implications of your findings.
9. What are the limitation of the present study?
10. Please explain how the product recommendation works for a user who (without user-ID) has not logged-in to the website?
11. Why authors used dimension reduction in the present study and how it effects the streaming learning data?
Minor comments:
1. Page 11, line 327 to 332: Please provide the reference to the claim for readers’ convenience.
2. Page 14, Equation 12: Please provide the reference to the claim for readers’ convenience.
Reviewer 2 Report
This paper extended one 7 pages paper to 27 pages with much content to consume in one single paper.
The paper has 10 sections with a mix of theoretical and mathematical as well as technical/systematic content to give a big picture of a complex system.
This ends with a lack of focus but filled with different levels of details and complexity so the readability is greatly degraded.
Authors are recommended to make revisions so that the paper is tightened up, consider whether every (esp those simple, or textbook style) single figure, table, fomula is really needed.
The paper should also highlight what are key take away messages.
Thee is no point to have third level sections titles such as 10.2.3 where there is only one two line paragraph.
Round 2
Reviewer 1 Report
The authors have addressed the suggested remarks. In sum: accepted.
Author Response
Thank you for your review.
Reviewer 2 Report
I still feel the paper is lacking a focus and major contribution point.
Round 3
Reviewer 2 Report
thanks for efforts to make the paper intact